# Extreme terahertz magnon multiplication induced by resonant magnetic pulse pairs

C. Huang [1,2,6], L. Luo[1,6], M. Mootz [1], J. Shang[3,4], P. Man[3,4], L. Su[3,4], I. E. Perakis [5], Y. X. Yao [1], A. Wu[3,4] & J. Wang [1,2] ✉

Nonlinear interactions of spin-waves and their quanta, magnons, have emerged as prominent candidates for interference-based technology, ranging from quantum transduction to antiferromagnetic spintronics. Yet magnon multiplication in the terahertz (THz) spectral region represents a major challenge. Intense, resonant magnetic fields from THz pulse-pairs with controllable phases and amplitudes enable high order THz magnon multiplication, distinct from non-resonant nonlinearities such as the high harmonic generation by below-band gap electric fields. Here, we demonstrate exceptionally high-order THz nonlinear magnonics. It manifests as 7th-order spin-wave-mixing and 6th harmonic magnon generation in an antiferromagnetic orthoferrite. We use THz two-dimensional coherent spectroscopy to achieve high-sensitivity detection of nonlinear magnon interactions up to six-magnon quanta in strongly-driven many-magnon correlated states. The high-order magnon multiplication, supported by classical and quantum spin simulations, elucidates the significance of four-fold magnetic anisotropy and Dzyaloshinskii-Moriya symmetry breaking. Moreover, our results shed light on the potential quantum fluctuation properties inherent in nonlinear magnons.

Recently, there is growing evidence that non-thermal control of magnetism by light is feasible[1–17]. The THz frequency magnons[18–24] is central for emergent spin-wave interference-based devices[25] at least 1000 times faster than the magneto-optical recording and device technologies that use thermal-magnetic excitations. Furthermore, THz control of magnetism is cross-cutting for coherent magnonics[10–12,26–28], quantum magnetism[1,9,29], antiferromagnetic (AFM) spintronics[2,3,26], and quantum computing[30].

Microwave spin-wave excitation has been shown to induce megahertz frequency magnon multiplication[31]. However, THz frequency magnon multiplication remains an as yet unobserved nonlinear process, in which multiple THz magnons can fuse together to form a high-order magnon coherence. Recent experiments have applied THz frequency, two-dimensional coherent nonlinear

spectroscopy (THz-2DCS) to quantum systems, such as low-dimensional semiconductors[32,33] and superconductors[34–36]. THz-2DCS study of magnetic materials has also yielded interesting results such as revealing magnon second harmonic generation[37], upconversion[38] and mixing[39]. Thus far the results obtained are explained by calculating low-order spin susceptibilities. We place emphasis on observing the higher-order nonlinearity and collectivity of THz magnons, supported by classical and quantum spin simulations.

To explore high-order magnon multiplication and control, an AFM orthoferrite is driven and probed by THz-2DCS. We use the magnetic fields **B**(t) of intense THz pulse-pair to resonantly induce collective spin dynamics and correlation, as illustrated in Fig. 1a. The excitation by such pulse-pair enables us to measure the phase of the AFM magnonic coherent nonlinear emission, in addition to the amplitude as in regular

[1]Ames National Laboratory, Ames, IA 50011, USA. [2]Department of Physics and Astronomy, Iowa State University, Ames, IA 50011, USA. [3]State Key Laboratory of High Performance Ceramics and Superfine Microstructure, Shanghai Institute of Ceramics, Chinese Academy of Sciences, Shanghai 201899, China. [4]Center of Materials Science and Optoelectronics Engineering, University of Chinese Academy of Sciences, Beijing 100049, China. [5]Department of Physics, University of Alabama at Birmingham, Birmingham, AL 35294-1170, USA. [6]These authors contributed equally: C. Huang, L. Luo. ✉e-mail: jgwang@iastate.edu; jgwang@ameslab.gov

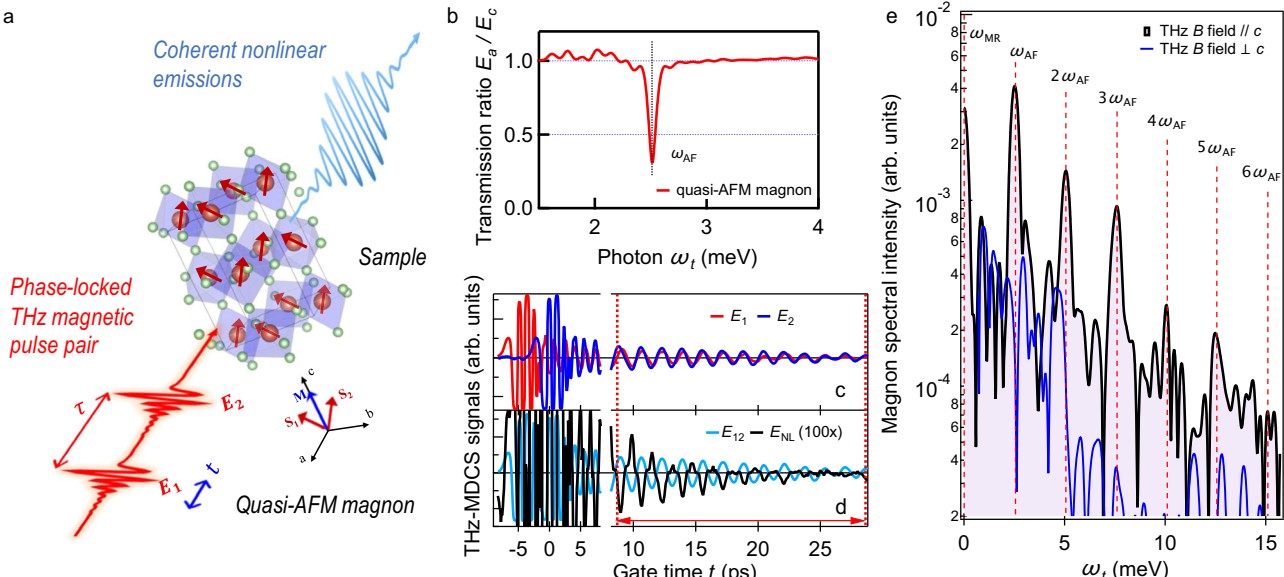

**Fig. 1 | THz-2DCS of magnonic multiplication in Sm$_{0.4}$Er$_{0.6}$FeO$_3$. a** Experimental schematics. Two phase-locked THz pulses with inter-pulse delay $\tau$ are focused onto the sample. The magnetic fields of this THz pulse-pair are polarized parallel to the sample $c$-axis to excite the quasi-AFM mode. The coherent nonlinear emission signals are detected through electro-optic sampling. **b** The static THz transmission measurement of the sample, which describes the linear magnonic response, shows a peak at the magnon mode energy ~ 2.5 meV ( ~ 0.6 THz). **c** and **d**, Typical four channel THz signals measured by THz-2DCS at a fixed delay time $\tau = 4.8$ ps. Two individual THz pulses used here, $E_1$ (red line) and $E_2$ (blue line), are shown in **c**. The coherent emission signal when both pulses are present, $E_{12}$ (light blue line), as well as the extracted nonlinear signal $E_{NL}$ (black line), obtained by subtracting the two individual pulse contributions from $E_{12}$ (zoom in 100 ×), are plotted in **d**. The Fourier Transform of $E_{NL}$ is performed for the time window after the pulse indicated by the two dotted red lines. **e** The $E_{NL}$ spectrum shows up to six-magnon multiplication (dashed lines) for THz magnetic fields parallel to the $c$-axis (black solid line). The magnon nonlinear peaks are absent for THz magnetic fields perpendicular to the $c$-axis (blue solid line).

pump-probe (PP) experiments. This approach opens the door to the possibility of driving high-order nonlinear wave-mixing, which is controlled by the interactions of multiple, long-lasting magnon excitations persisting well beyond the duration of the initial laser pulses. We emphasize three additional points here. First, by leveraging both the real-time and relative phase dependencies of coherent emission, we can effectively distinguish, in frequency space after Fourier Transform (FT), discrete spectral peaks resulting from magnon multiplications of varying orders. Second, it is important to distinguish the magnon multiplication from the conventional high harmonic generation of solids[40]. Magnon multiplication utilizes resonant excitations of collective modes leading to high harmonics that emerge after the pulses. In contrast, the conventional electric field-induced high harmonics utilize photoexcitation occurring in the transparency region which exists only during the period of laser excitation. Third, resonant THz photoexcitations have proven to be highly effective in sustaining long-lasting coherence of low-energy collective modes in quantum materials. This is evidenced by recent studies involving topological materials[41,42], semiconductors[43] and superconductors[36].

In this Letter, we demonstrate high-order nonlinear magnonics by performing THz-2DCS measurements of Sm$_{0.4}$Er$_{0.6}$FeO$_3$ using an intense, resonant THz laser pulse-pair. The sensitive detection of coherent spectra enables us to observe up to sixth harmonic generation (6HG) and seventh-order spin wave-mixing (7WM) signals, which significantly exceed what has been previously reported. These distinct magnon multiplication peaks demonstrate a "super" resolution tomography of long-lived interacting magnon quantum coherences and distinguish between different many-body spin correlation functions that describe strongly-driven magnon states. By simulating the coherent dynamics generated by large pulse-pair-driven spin deviations from the equilibrium magnetization orientation, we reproduce the experimental THz-2DCS spectra and identify multi-magnon interaction originating from the dynamical interplay

of four-fold magnetic anisotropy and Dzyaloshinskii-Moriya (DM) symmetry breaking.

## Results

Figure 1a illustrates our THz-2DCS measurement of quasi-AFM magnons in $a$-cut rare-earth orthoferrite sample Sm$_{0.4}$Er$_{0.6}$FeO$_3$ (Methods) at room temperature. This sample exhibits canted AFM order in equilibrium (red arrows), with a small net magnetization pointing along the $c$-axis (blue arrow). The measured linear THz response clearly shows a quasi-AFM magnon mode with energy at $\omega_{AF}$ ~ 2.5 meV (Fig. 1b). In our THz-2DCS experiment, two collinear THz pulses $E_1$ and $E_2$ (red lines) with a time delay $\tau$ (red arrow) are used, whose magnetic fields are polarized parallel to the sample $c$-axis. The transmitted nonlinear emission excited from such intense THz pulse-pair at a fixed $\tau$ is detected as the function of the gate time $t$ through electro-optical sampling.

Figure 1c presents representative time-domain traces of the two THz pulses, $E_1$ and $E_2$ (red and blue solid lines), with electric field strength up to 47.5 MV/m each. We clearly observe the THz light-driven single-order, long-lasting magnon quantum coherence. We expect that magnon-phonon interactions mainly drive the oscillation decay. Figure 1d shows the coherent emission trace when both pulses are present, $E_{12}$ (light blue line), and the extracted nonlinear signal, $E_{NL}(t,\tau) = E_{12}(t,\tau) - E_1(t,\tau) - E_2(t)$ (black line), recorded as the function of gate time $t$ for various inter-pulse delay times $\tau$. The exemplary $E_{NL}(t,\tau)$ at the fixed $\tau = 4.8$ ps in Fig. 1d is generated by the constructive interference of THz-excited magnon with energy ~ 2.5 meV measured in the static THz spectrum (Fig. 1b). This result illustrates the simultaneous amplitude- and phase-resolved detection of the coherent nonlinear response. The subtraction of the single pulse signals $E_1$ and $E_2$ from the total signal $E_{12}$ allows an extraction of pure coherent nonlinear emissions arising from magnonic nonlinear interactions. The measured long-time dynamics of $E_{NL}(t,\tau)$ exhibits pronounced, anharmonic oscillations lasting for a long time, as seen within the

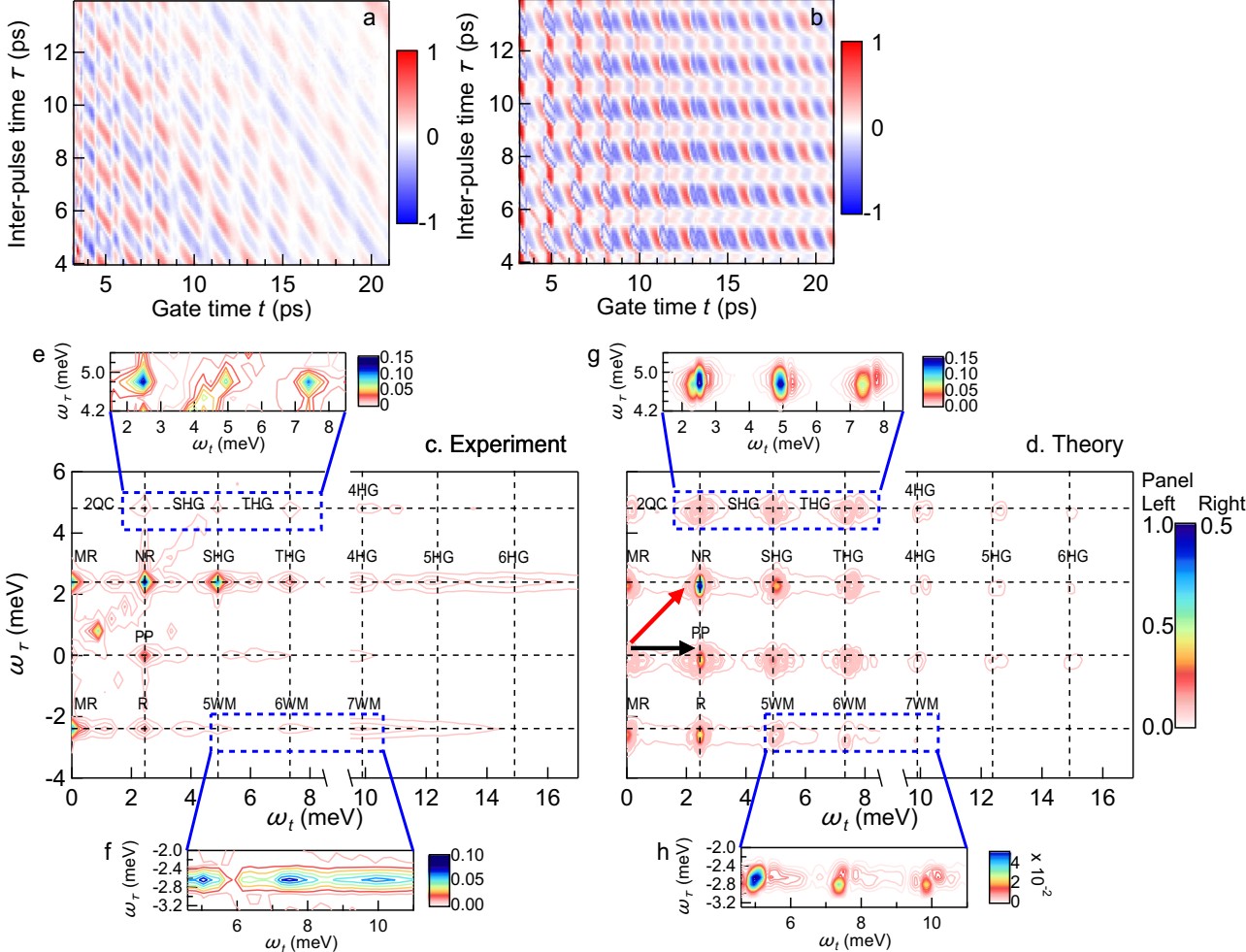

**Fig. 2 | Super-resolution tomography of magnonic multiplication and correlation. a** Two-dimensional false-color plot of the measured coherent nonlinear transmission signals $E_{NL}(t, \tau)$ as a function of the gate time $t$ and the delay $\tau$ induced with the electric field strength 47.5 MV/m. **b** The simulated $E_{NL}(t, \tau)$. **c** and **d** The $E_{NL}(\omega_t, \omega_\tau)$ spectra from 2D-FT of time domain images **a** and **b**. The THz-2DCS spectra show PP, R, NR, and 2QC peaks, which are generated by third-order nonlinear processes. Second-order nonlinear processes lead to MR and SHG peaks.

Importantly, strong THz driving also leads here to formation of high-order nonlinear signals including THG, 4HG, 5HG, 6HG as well as 5WM, 6WM and 7WM peaks. The frequency vectors for two incident pulses 1 and 2, $\omega_1 = (\omega_{AF}, \omega_{AF})$ and $\omega_2 = (\omega_{AF}, 0)$, are marked as red and black arrows. **e** and **f** Zoom-in view of new high harmonic and wave mixing peaks observed in our experiments which involve up to six magnon quanta (main text). **g** and **h** Corresponding zoom-in peaks reproduced by theory.

temporal range marked by a double-arrowed solid line in Fig. 1d. This observation indicates that nonlinear effects of strongly-driven AFM magnons last well beyond the pulse duration. Figure 1e shows a spectrum of $E_{NL}(t, \tau)$ obtained over the long oscillation period, which clearly demonstrates the presence of high-order magnonic nonlinearities. This spectrum shows sharp magnon multiplication peaks at frequencies up to six times the magnon energy (Fig. 1b), as marked by the dashed lines. They have been absent in prior THz-2DCS studies of ErFeO₃³⁷,³⁸. The harmonic peaks are robust from interactions among multiple long-lasting magnon quantum excitations, rather than laser fields. They appear for magnetic fields parallel to the *c*-axis (black line) and are absent for perpendicular to the *c*-axis (blue line).

Figure 2a and c show two-dimensional (2D) temporal and spectral profiles of the experimentally measured coherent nonlinear signals $E_{NL}(t, \tau)$. For comparison, Fig. 2b and d present the corresponding results of our numerical simulations, to be discussed below. The oscillations along the horizontal gate time *t*-axis represent the time evolution of magnon interaction for different pulse-pair time delays $\tau$. These gate time *t*-oscillations are pronounced even when two incident pulses do not overlap in time, e.g., for a long inter-pulse delay $\tau = 12$ ps. The observation of strong *t*-oscillations at

such large $\tau$ indicates that light-induced coherence is stored in the magnon system well after the laser pulse. The oscillatory time-dependent nonlinear signal along the $\tau$-axis for a fixed $t$ displays coherent enhancement due to constructive interference when $\tau$ corresponds to in-phase magnon excitation, while the signal diminishes due to destructive interference when $\tau$ corresponds to out-of-phase magnon excitation. Figure 2c shows the THz-2DCS spectra obtained by 2D-FT of $E_{NL}(t, \tau)$ with respect to gate time $t$ (frequency $\omega_t$) and inter-pulse delay $\tau$ (frequency $\omega_\tau$), where a series of discrete spectral peaks can be observed (Table 1).

Next, we will first discuss the peaks originating from conventional magnon rectification (MR), pump-probe (PP), and phase-reversal (R) or non-reversal (NR), followed by the newly discovered nonlinear magnonic wave-mixing and harmonic generation processes produced by high-order magnetic interactions as summarized in Table 1. Particularly, Fig. 2c clearly shows that strong THz field driving leads to the formation of high-order nonlinear peaks never seen before, including third-harmonic (THG), fourth-harmonic (4HG), fifth-harmonic (5HG), and sixth-harmonic generation (6HG) as well as five-wave (5WM), six-wave (6WM), and seven-wave mixing (7WM) peaks, as discussed below.

**Table 1 | Nonlinear magnon processes contributing to the THz-2DCS spectra**

| signal | nonlinear process | frequency space |
|---|---|---|
| MR | $\omega_1 - \omega_2$ | $(0, \omega_{AF})$ |
| MR | $\omega_2 - \omega_1$ | $(0, -\omega_{AF})$ |
| PP | $(\omega_1 - \omega_1) + \omega_2$ | $(\omega_{AF}, 0)$ |
| NR | $(\omega_1 - \omega_2) + \omega_2$ | $(\omega_{AF}, \omega_{AF})$ |
| 2QC | $2\omega_1 - \omega_2$ | $(\omega_{AF}, 2\omega_{AF})$ |
| R | $(\omega_2 - \omega_1) + \omega_2$ | $(\omega_{AF}, -\omega_{AF})$ |
| SHG | $2\omega_1$ | $(2\omega_{AF}, 2\omega_{AF})$ |
| SHG | $\omega_1 + \omega_2$ | $(2\omega_{AF}, \omega_{AF})$ |
| 5WM | $3\omega_2 - \omega_1$ | $(2\omega_{AF}, -\omega_{AF})$ |
| THG | $\omega_1 + 2\omega_2$ | $(3\omega_{AF}, \omega_{AF})$ |
| THG | $2\omega_1 + \omega_2$ | $(3\omega_{AF}, 2\omega_{AF})$ |
| 6WM | $4\omega_2 - \omega_1$ | $(3\omega_{AF}, -\omega_{AF})$ |
| 4HG | $3\omega_2 + \omega_1$ | $(4\omega_{AF}, \omega_{AF})$ |
| 4HG | $2\omega_2 + 2\omega_1$ | $(4\omega_{AF}, 2\omega_{AF})$ |
| 7WM | $5\omega_2 - \omega_1$ | $(4\omega_{AF}, -\omega_{AF})$ |
| 5HG | $4\omega_2 + \omega_1$ | $(5\omega_{AF}, \omega_{AF})$ |
| 6HG | $5\omega_2 + \omega_1$ | $(6\omega_{AF}, \omega_{AF})$ |

The different THz-2DCS peaks correspond to different nonlinear processes, which can be observed simultaneously and identified according to their locations in $(\omega_t, \omega_\tau)$ space. To analyze the measured 2D spectrum, we introduce frequency vectors for incident pulse 1 and 2 excitations, $\omega_1 = (\omega_{AF}, \omega_{AF})$ and $\omega_2 = (\omega_{AF}, 0)$ (red and black arrows in Fig. 2d), where $\omega_{AF} \sim 2.5$ meV denotes the AFM magnon mode energy. The several discrete peaks separated along the vertical axis (relative phase), as shown in Fig. 2c, demonstrate that the pulse-pair time delay $\tau$ controls phase coherence between magnons excited by two incident pulses. Figure 2e and f further show a zoom-in view of some high harmonic and wave mixing peaks observed in the experiment, while Fig. 2g and h show the corresponding peaks reproduced by the theory.

In order to compare the measured and simulated nonlinear processes, we first identify the physical origins of the series of discrete peaks seen in Fig. 2c along the vertical direction $\omega_t = \omega_{AF}$, located at $\omega_\tau = -\omega_{AF}$, 0, $\omega_{AF}$, and $2\omega_{AF}$. The $(\omega_{AF}, 0)$ peak arises from the PP process (Table 1) that reflects the dynamics of magnon coherent populations. In particular, two field interactions during pulse 1 create a magnon population via the difference frequency process $\omega_1 - \omega_1$. This population evolves during time $\tau$ and then interacts with a magnon excitation during pulse 2, resulting in a single magnon 1-quantum coherence (1QC). The peak at $(\omega_{AF}, 2\omega_{AF})$ shows the dynamics of a magnon 2-quantum coherence (2QC), which is generated by two magnon excitations during pulse 1 via the sum–frequency process $\omega_1 + \omega_1$. This 2-magnon coherence is probed through its interaction with a magnon excitation during pulse 2.

The two peaks located at $(\omega_{AF}, \omega_{AF})$ and $(\omega_{AF}, -\omega_{AF})$ arise from the interaction of a 1QC generated by pulse 1 and a 1QC generated by pulse 2. Their interaction creates a magnon population via the difference frequency processes $\pm(\omega_1 - \omega_2)$, whose interaction with a magnon excitation during pulse 2 leads to a 1QC which is either phase reversed (R) or not reversed (NR) with respect to the 1QC of pulse 1. We also observe peaks along $\omega_t = 3\omega_{AF}$ that correspond to third harmonic generation (THG) signals. Here, the interaction of a 2QC created by pulse 1(2) with a 1QC generated by pulse 2(1) produces a THG signal at $(3\omega_{AF}, 2\omega_{AF})$ and $(3\omega_{AF}, \omega_{AF})$.

The measured THz-2DCS spectra allow us to identify the role of the antisymmetric exchange DM interaction in the studied AFM system. This is possible through peaks generated by second-order

magnon nonlinear processes facilitated by the symmetry-breaking DM interaction. The latter interaction is already known to induce a canted AFM order in the ground state. Such spin canting with respect to the AFM orientation of a spin-up and a spin-down sublattice results in a net magnetization along the c-axis (blue arrow, Fig. 1a). The coherent spin dynamics along this axis then becomes anharmonic, which results in the second-order nonlinear peaks observed in the 2D spectra at $\omega_t = 2\omega_{AF}$ and at $\omega_t = 0$ (Fig. 2c) that involve two magnon excitations. The difference frequency processes $\omega_1 - \omega_2$ involving two magnon 1QC generate magnon rectification (MR) signals at $(0, \pm \omega_{AF})$ via the DM interaction. The latter antisymmetric exchange also generates second harmonic generation (SHG) peaks at $(2\omega_{AF}, \omega_{AF})$ and $(2\omega_{AF}, 2\omega_{AF})$. These SHG peaks arise from the sum-frequency generation processes $\omega_1 + \omega_2$ and $\omega_1 + \omega_1$, respectively.

Finally, the THz-2DCS spectra at Fig. 2c and e reveal high-order nonlinear peaks involving up to six magnon quanta. In particular, we observe strong 4HG, 5HG, and 6HG peaks, and also strong 5WM, 6WM, and 7WM nonlinear signals, shown in Fig. 2f. It is important to note that the 2D frequency resolution enabled by the THz-2DCS allows us to not only excite coherent magnons but also achieve a "super" resolution tomography different from the conventional high harmonic generation spectroscopies that only detect one dimensional spectra. In the 2D frequency plane, the same, high-order magnon harmonics will appear as different correlated THz peaks along multiple axes, e.g., 6HG and 5WM peaks separated in 2D space. Such strong correlation enables the deterministic assignment of higher–order nonlinear peaks, even in the presence of measurement noise or FT artifacts, which were very difficult to discern before. As discussed below, our numerical simulations shown in Fig. 2d, g and h further corroborate our discovery of the higher-order magnon multiplication peaks. We attribute these high-order nonlinear signals to the existence of both four-fold magnetic anisotropy and DM interaction in our AFM system. The origin of all observed THz-2DCS peaks is summarized in Table 1. Note that additional experimental signals at frequencies less than $\omega_{AF}$ are not captured by our mean field model. While we do not discuss such signals here, they may arise from parametric processes involving two magnons with finite momenta $\mathbf{q}$ and $-\mathbf{q}$ such that $\omega_{AF} = \omega_\mathbf{q} + \omega_{-\mathbf{q}}$[44].

To corroborate the above interpretation of experimental results, we use the following two sub-lattice Hamiltonian[45,46]:

$$H = J\mathbf{S}_1 \cdot \mathbf{S}_2 - \mathbf{D} \cdot (\mathbf{S}_1 \times \mathbf{S}_2) - \sum_{i=1}^{2}\left(K_a S_{i,a}^2 + K_c S_{i,c}^2\right)$$
$$- K_4 \sum_{i=1}^{2}\sum_{j=a,b,c} S_{i,j}^4 - \gamma\, \mathbf{B}(t) \cdot \sum_{i=1}^{2} \mathbf{S}_i. \tag{1}$$

This classical spin model effectively describes the AFM resonance mode and THz magnon multiplication behavior in orthoferrite systems where the DM interaction is the predominant canting mechanism[47]. The first term of Eq. (1) accounts for the AFM coupling between the neighboring spins $\mathbf{S}_1$ and $\mathbf{S}_2$ with exchange constant $J > 0$. The second term describes the Dzyaloshinskii-Moriya (DM) interaction, with antisymmetric exchange parameter $\mathbf{D}$ aligned along the b-axis. The orthorhombic crystalline anisotropy is described by the third and fourth terms of the Hamiltonian, with anisotropy constants $K_a$ and $K_c$ along the a and c axes, respectively. To explain the observed high-order nonlinearities, we include the four-fold anisotropy $K_4$. Finally, the driving of the spin system by the laser magnetic field $\mathbf{B}(t)$ is described by the Zeeman interaction (the last term of the above Hamiltonian), where $\gamma = g\mu_B/\hbar$ is the gyromagnetic ratio, $g = 2$ is the Landé g-factor, and $\mu_B$ denotes the Bohr magneton. Additional parameters used for classical spin simulations are provided in Table 2 for reference.

We have simulated the nonlinear spin dynamics by solving the full Landau-Lifshitz-Gilbert equations[2,37] (LLG, Eq. (2) in Methods) derived from the Hamiltonian Eq. (1). Non-interacting magnon excitations can be described by linearizing this LLG equation in the case of small amplitude oscillations around the equilibrium magnetization. This approximation is valid for weak $B(t)$. However, the full nonlinearities described by the LLG equation, central in many spin-wave device concepts, become critical for strong fields, which drive large amplitude of magnon oscillations. Here we characterize such magnon nonlinearities by solving the full nonlinear LLG equation (Eq. (2), Methods) for two magnetic fields polarized along sample $c$-axis. We obtain THz-2DCS spectra by doing 2D-FT of the nonlinear magnetization $M_{NL}$, which is obtained by subtracting the magnetizations induced by the two individual pulses from the nonlinear response of the net magnetization $|M|$ under intense pulse-pair excitation (Methods). Figure 2b and d present the 2D temporal profile $M_{NL}(t, \tau)$ and spectrum $M_{NL}(\omega_t, \omega_\tau)$, respectively. The simulated $M_{NL}(\omega_t, \omega_\tau)$ is fully consistent with the measured spectrum in Fig. 2c. This overall agreement of the two-time/two-frequency dependencies between theory and experiment allows us to assign the different discrete peaks observed in the experiment to specific nonlinear processes and magnetic interactions. The agreement also corroborates the magnonic origin of the observed high-order nonlinearities and symmetry-breaking peaks. Note that the remaining discrepancy with experimental results underscores the significance of considering the quantum properties of the spin in describing THz-2DCS experiments discussed in more detail below.

Figure 3a–f show the field dependence of the nonlinear spectral signals $E_{NL}(\omega_t, \tau)$ at magnonic multiplication frequencies $\omega_t =$ 0, 1, 2, 3, 4, 5$\omega_{AF}$. The $E_{NL}(\omega_t, \tau)$ traces are taken at $\tau =$ 4.8 ps to maximize the sensitivity similar to Fig. 1e. These high-order magnon multiplication peaks clearly grow nonlinearly with increasing field strength and follow power-law dependencies. By fitting (solid lines) the 5$\omega_{AF}$ peak amplitude in Fig. 3a, we see that is clearly scales as $E^5$. Note that 0$\omega_{AF}$ and 1$\omega_{AF}$ peaks scale as $E^2$ and $E^3$, respectively, confirming that they originate from second-order and third-order nonlinear processes as discussed above.

Figure 3g and h compare a pulse-pair and single pulse pumping for generating high-order magnon multiplication peaks at 2, 3, 4$\omega_{AF}$ (dashed vertical lines). The extracted $E_{NL}(\omega_t, \tau)$ spectra at fixed $\tau =$ 4.8 ps in Fig. 3g is presented for 100% (purple line), 70% (green line), and 50% (orange line) of the maximum THz field strength and grow nonlinearly with increasing field. Compared to the $E_{NL}(\omega_t, \tau)$ spectra, the single–pulse signal, $E_2(\omega_t)$ (Fig. 3h), does not show any significant high-order magnon multiplication peaks. Magnonic nonlinearity and interaction effects are hidden in the response to a single strong pulse, but become prominent in the THz-2DCS. In the latter case, the single-pulse responses are subtracted out, so the pure nonlinear signals are determined by correlations between two pulse excitations, rather than by single pulse nonlinearities. As a result, THz-MDCS allows us to resolve the nonlinear interactions between coherent magnons excited by different pulses and associate them with different peaks in 2D frequency space.

## Discussion

Figure 4 presents our numerical calculations in more detail. Particularly, it provides direct insight into the physical origin of the different THz-2DCS spectral peaks by comparing the contributions from different magnetic interactions and anisotropies. We compare the spectra $M_{NL}(\omega_t, \omega_\tau)$ calculated in an electric field strength of ~45 MV/m between (i) the full-term calculation (Fig. 4a), (ii) a calculation without DM interaction, $D = 0$, where the ground state has no spin canting

**Table 2 | Parameters used in the classical simulations**

| Parameter | value used |
|---|---|
| Exchange constant $J$ | 5.0 meV |
| Anisotropy constant $K_a$ | $1.2 \times 10^{-2}$ meV |
| Anisotropy constant $K_c$ | $0.46 \times 10^{-2}$ meV |
| Anisotropy constant $K_4$ | $1.8 \times 10^{-4}$ meV |
| Antisymmetric exchange constant $D$ | $7.6 \times 10^{-2}$ meV |
| Total spin number $S$ | 5/2 |
| $\alpha$ | $1.3 \times 10^{-3}$ |

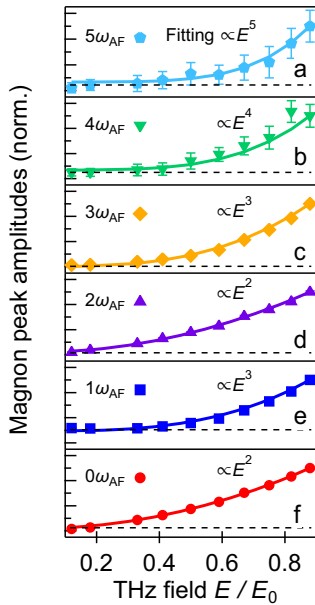

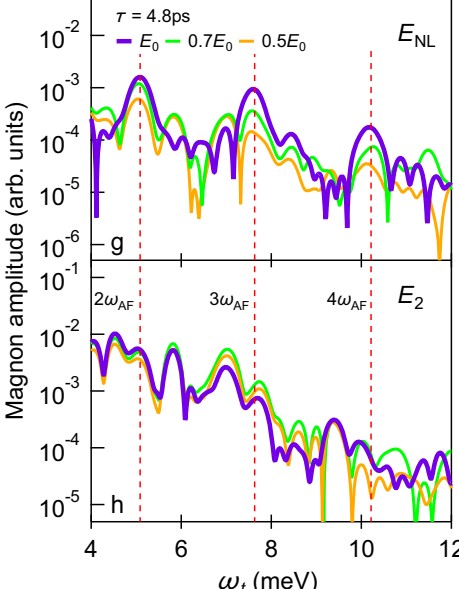

**Fig. 3 | THz field dependence of magnonic multiplication. a–f** Magnonic multiplication peak amplitudes as a function of THz fields by recording spectral signals $E_{NL}(\omega_t, \tau)$ (Fig. 1e) at frequencies of 0, 1, 2, 3, 4, 5$\omega_{AF}$ and inter-pulse delay $\tau =$ 4.8 ps. Solid lines are power-law fitting of the experimental data (markers). The error bars marked in **a–f** are obtained by averaging background noise following the same data taking protocol and fall within the marker sizes for the lower orders. **g** $E_{NL}(\omega_t, \tau)$ spectra at fixed inter-pulse delay $\tau =$ 4.8 ps for 100% (purple line), 70% (green line), and 50% (orange line) of THz field excitations. **h** Corresponding single pulse spectra $E_2(\omega_t)$ at the same field strengths.

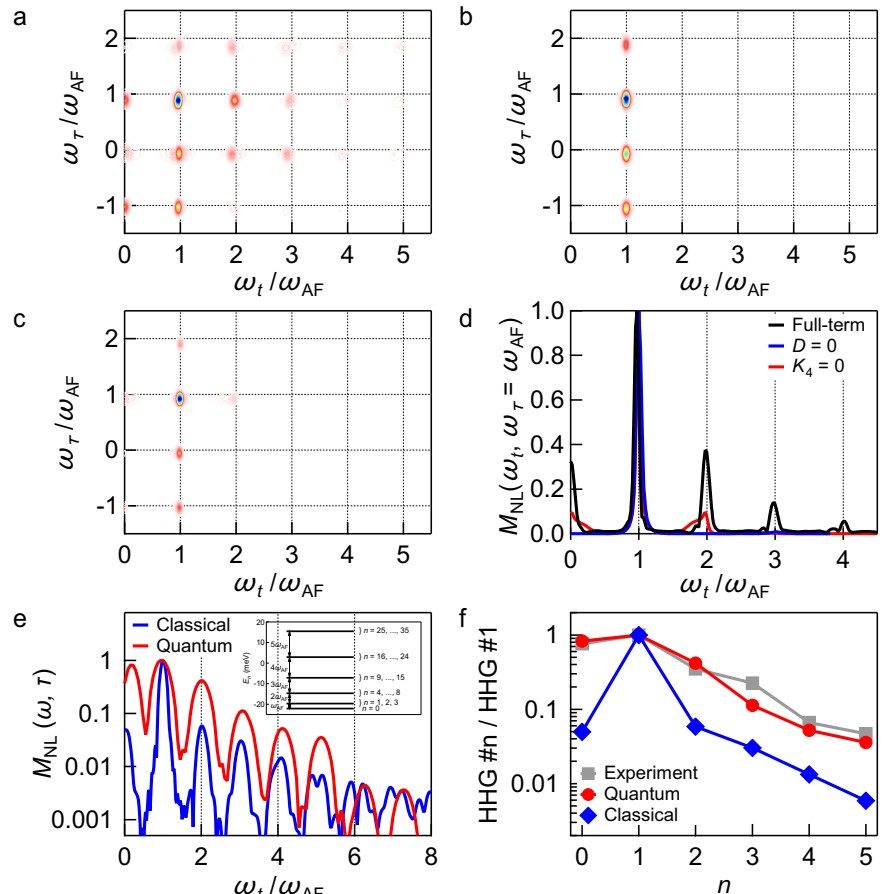

**Fig. 4 | Simulations of high-order nonlinear magnonic correlations.** $M_{NL}(\omega_t, \omega_\tau)$ for **a** the full-term calculation, **b** a calculation without DM interaction, $D = 0$, where the ground state has no spin canting, i. e., $M = 0$ in the ground state, and **c** a calculation without four-fold anisotropy in the Hamiltonian, $K_4 = 0$. **d** $M_{NL}(\omega_t, \omega_\tau)$ at fixed $\omega_\tau = \omega_{AF}$. The full-term calculation (black line) is compared with simulations where the DM interaction is switched off (blue line) and the four-fold magnetic anisotropy is switched off (red line). **e** $M^c_{NL}(\omega_t, \tau)$ at a fixed inter-pulse delay of $\tau = 4.8$ ps. The result of the full quantum spin simulation (red line) is shown together

with the result of the classical simulation (blue line). The quantum spin simulation yields stronger high-harmonic generation peaks (vertical dashed lines) compared to the classical simulation. Inset: Eigenenergies $E_n$ of the unperturbed quantum spin Hamiltonian which has $(2s+1)^2 = 36$ eigenstates for the studied two-site spin-$s = 5/2$ system. **f** Ratio between the strength of $n$th harmonic and fundamental harmonic generation peaks. The ratios extracted from the experimental mHHG spectrum in Fig. 1e (squares) are compared with the results of the quantum spin simulation (circles) and the classical simulation (rectangles).

(Fig. 4b), and (iii) a calculation without four-fold magnetic anisotropy, i. e., by setting $K_4 = 0$ in the Hamiltonian (1) (Fig. 4c). All even-order wave-mixing signals vanish when the DM interaction is switched off, consistent with earlier works[37]. In addition, the third and higher odd-order nonlinear magnon signals are strongly suppressed (Fig. 4b). The third- and higher-order nonlinear magnon signals are also strongly suppressed if $K_4 = 0$ (Fig. 4c). The suppression of high-order signals in the simulations (Fig. 4b and c) indicates that the high-order nonlinear peaks observed in THz-2DCS experiment are generated by the four-fold magnetic anisotropy as well as the DM interaction. This interpretation can be seen more clearly by plotting a spectral slice along $\omega_\tau = \omega_{AF}$ in Fig. 4d, where the full-term calculation (black line) is compared with simulations where (i) the DM interaction is switched off (blue line), or (ii) the four-fold magnetic anisotropy is switched off (red line). This observation reveals the existence of a four-fold magnetic anisotropy and also demonstrates that the high-order magnonic nonlinearities arise from the dynamical interplay between DM interaction and four-fold magnetic anisotropy.

Figure 4e and f underscore the significance of considering the quantum fluctuation properties of spins in describing the THz-2DCS spectra. While the simulations based on the classical spin LLG model in Fig. 2d can replicate the positions of the peaks in the THz-2DCS spectra, the strengths of the high-order nonlinear signals using

classical spins are much weaker compared to the experimental data in Figs. 1e and 2c. For instance, the experimental data (gray squares) at $\omega_\tau = 2\omega_{AF}$ and $\omega_\tau = 3\omega_{AF}$ exhibits substantially larger high-harmonic generation signals in Fig. 4f compared to the result of the classical simulations (blue diamonds). To understand this we have also calculated $\mathbf{M}_{NL}(\omega_t, \omega_\tau)$ for the quantum spin version of Hamiltonian (1)[48]. This is done by replacing the classical spins $\mathbf{S}_i$ in Eq. (1) by quantum spin operators $\hat{\mathbf{S}}_i$ at site $i$ of spin magnitude $S = 5/2$. The resulting two-site quantum spin Hamiltonian has $(2S+1)^2 = 36$ eigenstates $|\Psi_n\rangle$ and eigenenergies $E_n$. The inset of Fig. 4e shows the eigenenergy spectrum of the quantum spin model. We have used the same parameters for $K_a$, $K_c$, and $K_4$ as in the classical simulations and adjusted $J$ and $D$ such that the magnon frequency, defined by $\omega_{AF} = E_1 - E_0$ in the quantum spin model, matches the one observed in the experiment (Table 3). The energy spectrum is anharmonic, however, the difference between the eigenenergies is given by multiples of the magnon energys such that one expects a harmonic energy spectrum.

To model the transmitted magnetic field, we calculate the dynamics of the magnetization along $c$-direction, $M^c(t) \equiv \langle\Psi(t)|(\hat{S}_{1,c} + \hat{S}_{2,c})|\Psi(t)\rangle$. Here $|\Psi(t)\rangle$ is the quantum state at time $t$ which is obtained via exact diagonalization. The calculated nonlinear magnetization $M^c_{NL}(t,\tau) = M^c_{12}(t,\tau) - M^c_1(t,\tau) - M^c_2(t)$ is presented in Fig. 4e (red line) at a fixed inter-pulse delay of $\tau = 4.8$ ps as in Fig. 1e. We

**Table 3 | Parameters used in the quantum spin simulations**

| Parameter | value used |
|---|---|
| Exchange constant $J$ | 2.5 meV |
| Anisotropy constant $K_a$ | $1.2 \times 10^{-2}$ meV |
| Anisotropy constant $K_c$ | $0.46 \times 10^{-2}$ meV |
| Anisotropy constant $K_4$ | $1.8 \times 10^{-4}$ meV |
| Antisymmetric exchange constant $D$ | 0.5 meV |
| Total spin number $S$ | 5/2 |

observe high-harmonic generation peaks (vertical dashed lines) up to seventh-order. Compared to that, the result of the corresponding classical simulation in Fig. 4e (blue line) exhibits about one-order of magnitude smaller second to fifth-order harmonic generation signals. To identify the origin of the strong magnonic high-harmonic generation signals of the quantum spin simulation, we express $|\Psi(t)\rangle$ after the magnetic field pulse-pair excitation in terms of the eigenstates $|\Psi_n\rangle$, yielding $|\Psi(t)\rangle = \sum_n a_n e^{-iE_n t} |\Psi_n\rangle$ with $a_n = \langle \Psi_n | \Psi(t=0)\rangle$. The dynamics of magnetization can then be written as $M^c(t) = \sum_{j,k} M_{j,k}^c a_j^* a_k e^{-i(E_k - E_j)t}$ with magnetic dipole matrix elements $M_{j,k}^c = \langle \Psi_j | (\hat{S}_{1,c} + \hat{S}_{2,c}) | \Psi_k\rangle$. As a result, the peaks in the spectrum arise from transitions between different eigenstates. The contribution of the transitions between the different eigenstates to $M_c(t)$ is determined by strength of the magnetic dipole matrix elements $M_{j,k}^c$. Since the difference between eigenenergies (inset of Fig. 4e) corresponds to multiples of the magnon frequency, we obtain a harmonic $M^c$ spectrum as observed in Fig. 4e.

Figure 4f directly compares the magnonic high harmonic generation observed in the simulations and experiment by plotting the ratio between the peak strengths of the $n$th harmonic and the fundamental harmonic. The result of the experiment (gray squares) is compared with the ratios deduced from the quantum-spin simulation (red circles) and classical simulation (blue diamonds). The quantum spin simulation agrees well with the ratios from experiment while the classical simulation produces substantially smaller high-harmonic generation peaks. Our results from the quantum spin calculation depicted in Fig. 4f emphasize the importance of further investigation into the pronounced higher-order magnon nonlinearities observed in our THz-2DCS experiment, which notably exceed the predictions of the classical spin model. Additional factors also need to be considered in the future quantitative analysis of high-order magnon nonlinear peaks, including the THz electro-optic sampling response functions[49] and THz propagation effects[50,51]. But we expect neither of these effects to substantially modify the peak strengths of the $2\omega_{AF}$ and $3\omega_{AF}$ nonlinear magnon peaks, which constitute the focus of the quantum spin calculations.

At last, we elaborate three key distinctions of the THz magnon multiplication observed in this work. First, magnon multiplication are generated by utilizing resonant excitation of collective spin wave modes with long-lasting quantum coherence that persists well after the laser pulses. High harmonic generation (HHG) in solids often occurs during the period of laser excitation. Second, THz-2DCS achieves super-resolution tomography of magnon interaction and nonlinearity. Our results reveal four-fold magnetic anisotropy and the Dzyaloshinskii–Moriya interaction in creating interactions among multiple long-lasting magnon quantum excitations. Such interaction leads to the high-order magnon multiplication. Third, the magnon multiplication peaks in the 2D spectrum can arise from resonant transitions between different quantum spin eigenstates[52]. Consequently, magnon multiplication nonlinearity can potentially provide direct information about quantum spin systems. Our results provide compelling implications of considering the quantum fluctuation properties of spins.

In summary, we demonstrate the extremely high-order magnonic multiplication and wave-mixing at THz frequency by driving a canted antiferromagnetic state. Our experimental results are well reproduced by both classical and quantum spin simulations, signifying the importance of four-fold magnetic anisotropy and DM symmetry breaking, also the fluctuation properties of quantum spins in the higher order magnon multiplication. Our demonstration of magnonic nonlinearity and collectivity through THz coherent spectroscopy can be further extended to pursue nonlinear quantum magnonics in parallel with nonlinear quantum optics.

## Methods
### Sample
Polycrystalline sample $Sm_{0.4}Er_{0.6}FeO_3$ was prepared by the conventional solid state reaction method, using the high purity oxide powder $Sm_2O_3$ (99.99%), $Er_2O_3$ (99.99%) and $Fe_2O_3$ (99.99%) as the starting materials. The mixture was pressed into pellets and sintered in air at 1400 K for 20 h. After confirmation by X-ray diffraction that the products had converted to the orthoferrite structure, the powder was then compacted into the rods. The rods were sintered in a vertical tube furnace at 1500 K for 15 h in air. Single crystal growth was performed in the optical floating-zone furnace (FZ-T-10000-H-VI-P-SH, Crystal System Corp).

The sample measured in the experiment is $a$-cut $Sm_{0.4}Er_{0.6}FeO_3$ single crystal with size around 10 mm × 5 mm × 1 mm. At room temperature, the canted spin antiferromagnetic order leads to a small net magnetization along $c$-axis (Fig. 1a). The magnetic resonance mode (quasi-AFM mode) can be excited if $\mathbf{B}_{THz}$ is parallel to this net magnetization. The in-plane axis is determined by rotating the sample to achieve the strongest magnon oscillation signal. In the experiment, we fixed sample $c$-axis perpendicular to THz electric field.

### THz two-dimensional coherent nonlinear spectroscopy
To generate and characterize THz-2DCS spectra, the $Sm_{0.4}Er_{0.6}FeO_3$ single crystal sample was excited by two broadband THz laser pulses. The measured coherent differential transmission $E_{NL}(t, \tau) = E_{12}(t, \tau) - E_1(t, \tau) - E_2(t)$ is plotted as a function of the gate (real) time $t$ and the delay time between pulses 1 and 2, $\tau$, which controls the relative phase accumulation between the two phase-locked THz fields. The time-resolved coherent nonlinear dynamics is then explored by varying the inter-pulse delay $\tau$. By measuring the electric fields in the time-domain through electro-optic sampling (EOS) by a third pulse, we achieve phase-resolved detection of the sample response as a function of gate time $t$ for each time delay $\tau$. The nonlinear response signals, separated from the linear response, leads to enhanced resolution that enables us to observe high-order magnonic nonlinearity.

### Experimental details
Our THz-2DCS setup is driven by a 800 nm Ti: Sapphire amplifier with 4 mJ pulse energy, 40 fs pulse duration, and 1 kHz repetition rate. The majority part of laser output (2.7 W) is split into two beams with similar power (1.35 W each), and the inter-pulse delay between them is controlled by a mechanical delay stage. Two beams are recombined and focused into a MgO doped $LiNbO_3$ crystal to generate intense THz pulses through the tilted pulse front scheme. The intense THz pulse has a peak field strength ~ 47.5 MV/m, central frequency ~ 1 THz (4.1 meV), and bandwidth ~ 1.5 THz[53,54]. The focused THz spot size at the sample position is ~ 1.3 mm. The rest part of the laser output is used to detect THz fields in time-domain by EOS using a 2 mm thick (110) oriented ZnTe crystal.

To acquire the THz nonlinear signal, a double modulation scheme has been used. Two optical choppers modulate two intense THz beams at a frequency of 500 Hz (pulse 1) and 250 Hz (pulse 2), respectively. Such a scheme allows to isolate nonlinear THz signals. The FT is performed at 7–28 ps range of the long lasting time-domain oscillatory traces as shown in Fig. 1c and d. The range is chosen to ensure avoidance of the stimulating THz pulses and echo pulses. The high-order THz magnon multiplication obtained was possible by (1)

choosing the Sm-doped magnetic material with the bigger four-fold magnetic anisotropy compared to un-doped ones, (2) applying an experimental scheme of 2 pulses vs. 1 pulse driving and (3) optimizing the detection sensitivity of 2D coherent nonlinear signals.

## THz–2DCS signal simulations

To simulate the THz–2DCS signals, we solve Landau–Lifshitz–Gilbert (LLG) equations derived from the Hamiltonian (1)[37]:

$$\frac{d\mathbf{S}_i}{dt} = \frac{\gamma}{1+\alpha^2}\left[\mathbf{S}_i \times \mathbf{B}_i^{\text{eff}}(t) + \frac{\alpha}{|\mathbf{S}_i|}\mathbf{S}_i \times \left(\mathbf{S}_i \times \mathbf{B}_i^{\text{eff}}(t)\right)\right], \qquad (2)$$

which describe the nonlinear coherent spin dynamics induced by the magnetic field $\mathbf{B}(t)$. Here, $\mathbf{B}_i^{\text{eff}}(t) = -\frac{1}{\gamma}\frac{\partial H}{\partial \mathbf{S}_i}$ corresponds to the time-dependent effective local magnetic field at site $i$ and $\alpha$ is the Gilbert damping coefficient. We numerically solve the full nonlinear LLG Eq. (2) with input THz magnetic field pulses 1 and 2 polarized along $c$-axis for realistic orthoferrite material parameters summarized in Table 2. The parameters used match the magnon mode frequency, free-induction decay, and nonlinear magnon signals observed in the experiment. We then calculate the net magnetization $\mathbf{M} \propto \mathbf{S}_1 + \mathbf{S}_2$ of the two AFM sub-lattices. To directly simulate our THz-2DCS experiment, we calculate the nonlinear differential magnetization $\mathbf{M}_{\text{NL}}$, which corresponds to the measured $\mathbf{E}_{\text{NL}}$. $\mathbf{M}_{\text{NL}}$ is obtained by computing the net magnetization induced by both pulses, $\mathbf{M}_{12}(t,\tau)$, as a function of the gate time $t$ and the inter-pulse time $\tau$, as well as the net magnetizations resulting from pulse 1, $\mathbf{M}_1(t,\tau)$, and pulse 2, $\mathbf{M}_2(t)$. The nonlinear differential magnetization is then given by $\mathbf{M}_{\text{NL}}(t,\tau) = \mathbf{M}_{12}(t,\tau) - \mathbf{M}_1(t,\tau) - \mathbf{M}_2(t)$ for the collinear geometry used in the experiment. The THz-2DCS spectra are obtained by Fourier Transform of $\mathbf{M}_{\text{NL}}(t,\tau)$ with respect to both $t$ (frequency $\omega_t$) and $\tau$ (frequency $\omega_\tau$). To analyze these 2D spectra, we introduce the frequency vector $(\omega_t, \omega_\tau)$. Since both pulses resonantly drive magnon 1-quantum coherences (1QC) characterized by the magnon frequency $\omega_{\text{AF}} \sim 2.5$ meV (Fig. 1b), the frequency vectors of the two pulses 1 and 2 can be written as $\omega_1 = (\omega_{\text{AF}}, \omega_{\text{AF}})$ and $\omega_2 = (\omega_{\text{AF}}, 0)$. The observed spectral peaks and corresponding nonlinear processes are summarized in Table 1.

## Data availability

All the data generated in this study are provided in the Source Data file. Source data are provided with this paper.

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

## Acknowledgements

This work was supported by the Ames National Laboratory, the US Department of Energy, Office of Science, Basic Energy Sciences, Materials Science and Engineering Division under contract No. DEAC0207CH11358 (project design and THz-2DCS spectroscopy). Synthesis and basic characterizations of crystals was supported by National Natural Science Foundation of China (NSFC, No.52272014), Science and Technology Commission of Shanghai Municipality (No. 21520711300) and CAS Project for Young Scientists in Basic Research (YSBR-024). The quantum spin model simulations were supported by the U.S. Department of Energy, Office of Science, National Quantum Information Science Research Centers, Co-design Center for Quantum Advantage (C2QA) under contract number DE-SC0012704.

## Author contributions

C.H. and L.L. performed the THz spectroscopy measurements with J.W.'s supervision. M.M., and Y.Y simulated the THz magnon nonlinearity using the quantum spin model. M.M., and I.E.P analyzed the THz spectra using the classical spin model. J.S., P.M., L.S. and A.W. grew the samples and performed crystalline quality and other basic characterizations. The paper is written by J.W., C.H., and M.M. with discussions from all authors. J.W. conceived and coordinated the project.

## Competing interests

The authors declare no competing interests.
