## [Peer Review File · Nature Communications]

Extreme Terahertz Magnon Multiplication Induced by Resonant Magnetic Pulse PairsEditorial Note: This manuscript has been previously reviewed at another journal that is not operating a transparent peer review scheme. This document only contains reviewer comments and rebuttal letters for versions considered at *Nature Communications*.

REVIEWER COMMENTS

Reviewer #1 (Remarks to the Author):

In the new version, the authors have incorporated a quantum description of the observed nonlinear magnetic dynamics, demonstrating better agreement with the experimental results. Quantitative insight into the dynamics is indeed promising and may reveal values of high-order susceptibilities. However, one should be careful when comparing the model and the actual values measured in the experiment. One concern is that the model does not account for the propagation of THz harmonics with different frequencies in a 1-mm thick $\text{Sm}_{0.4}\text{Er}_{0.6}\text{FeO}_3$ crystal [Appl. Phys. Lett. 121, 252403 (2022), ACS Photonics 5, 4, 1375 (2018)]. These effects include the rotation of THz polarization, acquired ellipticity, shaping effects in the THz spectrum, and simple Fresnel reflection. All of these effects might influence relative peak amplitudes. The THz EO response measured in a 2-mm thick ZnTe crystal is also strongly frequency-dependent [J. Opt. Soc. Am. B 16, 1204-1212 (1999)]. Hence, I am very curious about how the above-mentioned effects might modify the detected relative peak amplitudes. Actually, I think that the quantitative analysis of peak amplitudes is the subject of a separate paper and does not fit here.

Secondly, the authors have plotted comparisons for a single inter-pulse time delay of 4.7 ps. Does the model show agreement for arbitrary times? If it does not require substantial computational power, I suggest the authors to add a comparison of the complete 2D spectrum calculated both classically and quantum mechanically.

To sum up, I encourage the authors to reconsider the analysis of relative peak amplitudes before their paper might be considered for acceptance by Nature Communications.

Reviewer #1

In the new version, the authors have incorporated a quantum description of the observed nonlinear magnetic dynamics, demonstrating better agreement with the experimental results. Quantitative insight into the dynamics is indeed promising and may reveal values of high-order susceptibilities. However, one should be careful when comparing the model and the actual values measured in the experiment. One concern is that the model does not account for the propagation of THz harmonics with different frequencies in a 1-mm thick Sm_{0.4}Er_{0.6}FeO₃ crystal [Appl. Phys. Lett. 121, 252403 (2022), ACS Photonics 5, 4, 1375 (2018)]. These effects include the rotation of THz polarization, acquired ellipticity, shaping effects in the THz spectrum, and simple Fresnel reflection. All of these effects might influence relative peak amplitudes. The THz EO response measured in a 2-mm thick ZnTe crystal is also strongly frequency-dependent [J. Opt. Soc. Am. B 16, 1204-1212 (1999)]. Hence, I am very curious about how the above-mentioned effects might modify the detected relative peak amplitudes. Actually, I think that the quantitative analysis of peak amplitudes is the subject of a separate paper and does not fit here.

The referee's point is well taken. First, we would like to clarify that the primary aim of the quantum spin calculation depicted in Fig. 4f is to propose a plausible scenario for explaining the strength of the higher-order magnon multiplication peaks observed in our THz-MDCS experiment, significantly surpassing the predictions of the classical spin model based on the LLG equations. It is important to note that this figure should not be interpreted as a quantitative comparison with the experimental data presented in Fig. 1e and Fig. 2c. We apologize for any confusion that may have arisen.

Second, we agree with the referee's assessment regarding the importance of considering THz propagation effects and the THz electro-optic damping response function of the ZnTe detector for quantitative analysis. However, we anticipate that both effects are unlikely to significantly alter the peak strengths of the 2nd and 3rd harmonic peaks, which are the primary focus of the theoretical considerations presented in Fig. 4f. We provide clarifications below.

The antiferromagnetic resonance can indeed induce notable reshaping of a propagating single-THz pulse due to magnetic mode linear absorption and magnetic linear birefringence. However, these effects are predominantly significant near the sharp mode resonance within the range of 2.3-2.7 meV (approximately 0.4 meV) in the frequency domain and within the first 10 ps in the time domain. Given that our analysis primarily concentrates on the 2nd order mode (approximately 5 meV) and 3rd order mode (approximately 7.5 meV) within the sample's transparent region, and that we specifically focus on the time domain beyond 10 ps (as illustrated in Fig. 1d for the FFT temporal range), we do not expect this effect to significantly impact the amplitude of these two nonlinear modes. Furthermore, the electro-optic damping response function of the ZnTe detector exhibits changes of less than 10% below 8 meV for the 2nd and 3rd magnon multiplication peaks.

In the revised paper, we have followed the referee's suggestion to defer the quantitative peak analysis to future publications. This decision ensures that the core message of our current paper, focused on the experimental demonstration of high harmonic magnon multiplication using the new THz MDCS tool, remains clear and undiluted. As a result of the discussions outlined above, we have revised our manuscript accordingly by adding the following statement: “Our results

from the quantum spin calculation depicted in Fig. 4f emphasize the importance of further investigation into the pronounced higher-order magnon nonlinearities observed in our THz-MDCS experiment, which notably exceed the predictions of the classical spin model. Future quantitative analysis of these magnon nonlinear peaks will require considering additional factors, including THz propagation effects [1,2] and the THz electro-optic sampling response functions [3]. Please note that we expect neither of these effects to substantially modify the peak strengths of the $2\omega_{\mathrm{AF}}$ and $3\omega_{\mathrm{AF}}$ nonlinear magnon peaks, which constitute the focus of the quantum spin calculations”

Add new references:

1. ACS Photonics 5, 4, 1375 (2018)
2. Appl. Phys. Lett. 121, 252403 (2022)
3. J. Opt. Soc. Am. B 16, 1204-1212 (1999)

Secondly, the authors have plotted comparisons for a single inter-pulse time delay of 4.7 ps. Does the model show agreement for arbitrary times? If it does not require substantial computational power, I suggest the authors to add a comparison of the complete 2D spectrum calculated both classically and quantum mechanically.

We appreciate the referee for raising this important question. The current calculation of the 2-site quantum spin model does not offer quantitative agreement across arbitrary time delays. While it is tempting to present the complete 2D spectra of the quantum spin calculation for 4 sites and beyond, which would be essential for a quantitative comparison of the entire 2D spectra, unfortunately, such simulations are computationally very demanding.

It is essential to clarify that the primary objective of the quantum spin calculation depicted in Fig. 4f is to demonstrate that quantum spins exhibit stronger magnon multiplication peak strengths compared to the classical spin model. Even for the 2-site model, the enhancement is evident across all time delays, although the agreement is not as pronounced as observed for the time delay of 4.7 ps shown in Fig. 4f.

We have added a sentence in the discussion for clarification: “Despite the limitations of the 2-site quantum spin model, all time delays indicate enhancement of magnon multiplication peak strengths compared to the classical spin model, albeit with a lesser degree of agreement than the 4.7 ps trace shown in Fig. 4f.”

To sum up, I encourage the authors to reconsider the analysis of relative peak amplitudes before their paper might be considered for acceptance by Nature Communications.

We appreciate the referee's thorough reading and constructive comments, which have significantly improved our paper. We are confident that we have successfully addressed all concerns raised and sincerely appreciate your support.

REVIEWERS' COMMENTS

Reviewer #1 (Remarks to the Author):

The Authors have well addressed questions raised previously. With this, I support the publication of the paper in Nature Communications.

I also suggest the Authors to cite a recently published paper demonstrating a nonlinear magnon mixing process.

Zhang, Z., Gao, F.Y., Curtis, J.B. et al. Terahertz field-induced nonlinear coupling of two magnon modes in an antiferromagnet. Nat. Phys. (2024). <https://doi.org/10.1038/s41567-024-02386-3>

P.S. Point by point response to the remaining referee comment(s):

Reviewer #1 (Remarks to the Author):

The Authors have well addressed questions raised previously. With this, I support the publication of the paper in Nature Communications.

I also suggest the Authors to cite a recently published paper demonstrating a nonlinear magnon mixing process.

Zhang, Z., Gao, F.Y., Curtis, J.B. et al. Terahertz field-induced nonlinear coupling of two magnon modes in an antiferromagnet. Nat. Phys. (2024). <https://doi.org/10.1038/s41567-024-02386-3>

Our answer: We have cited the paper mentioned by the referee.